# Microbiology and Epidemiology of *Escherichia albertii*—An Emerging Elusive Foodborne Pathogen

**DOI:** 10.3390/microorganisms10050875

**Published:** 2022-04-22

**Authors:** Francis Muchaamba, Karen Barmettler, Andrea Treier, Kurt Houf, Roger Stephan

**Affiliations:** 1Institute for Food Safety and Hygiene, Vetsuisse Faculty, University of Zurich, 8057 Zurich, Switzerland; karen.barmettler@uzh.ch (K.B.); andreapatricia.treier@uzh.ch (A.T.); roger.stephan@uzh.ch (R.S.); 2Department of Veterinary and Biosciences, Faculty of Veterinary Medicine, Ghent University, 9820 Merelbeke, Belgium; kurt.houf@ugent.be

**Keywords:** *Escherichia albertii*, enteropathogen, diarrhea, isolation, misidentification, epidemiology

## Abstract

*Escherichia albertii*, a close relative of *E. coli*, is an emerging zoonotic foodborne pathogen associated with watery diarrhea mainly in children and immunocompromised individuals. *E. albertii* was initially classified as *eae*-positive *Hafnia alvei*, however, as more genetic and biochemical information became available it was reassigned to its current novel taxonomy. Its infections are common under conditions of poor hygiene with confirmed transmission via contaminated water and food, mainly poultry-based products. This pathogen has been isolated from various domestic and wild animals, with most isolates being derived from birds, implying that birds among other wild animals might act as its reservoir. Due to the absence of standardized isolation and identification protocols, *E. albertii* can be misidentified as other *Enterobacteriaceae*. Exploiting phenotypes such as its inability to ferment rhamnose and xylose and PCR assays targeting *E. albertii*-specific genes such as the cytolethal distending toxin and the DNA-binding transcriptional activator of cysteine biosynthesis encoding genes can be used to accurately identify this pathogen. Several gaps exist in our knowledge of *E. albertii* and need to be bridged. A deeper understanding of *E. albertii* epidemiology and physiology is required to allow the development of effective measures to control its transmission and infections. Overall, current data suggest that *E. albertii* might play a more significant role in global infectious diarrhea cases than previously assumed and is often overlooked or misidentified. Therefore, simple, and efficient diagnostic tools that cover *E. albertii* biodiversity are required for effective isolation and identification of this elusive agent of diarrhea.

## 1. Introduction

The emerging zoonotic foodborne pathogen *Escherichia albertii* is a Gram-negative rod, nonmotile, non-spore-forming, facultative anaerobe belonging to the *Enterobacteriaceae* family [1,2]. It is one of five species within the genus *Escherichia* [3]. Emerging data indicate that *E. albertii* is a diverse pathogen with two clades divided into eight genetic lineages, that have a geographical distribution bias, described to date [4,5,6]. *E. albertii* was first identified and reported as *Hafnia alvei* in diarrhea cases in children from Bangladesh and later reclassified as a new *Escherichia* species in 2003 [1,7,8]. Since then, *E. albertii* has been implicated in several outbreaks but is often mistakenly identified as enteropathogenic *E. coli* (EPEC) or enterohemorrhagic *E. coli* (EHEC) because of its genetic and phenotypic similarity to these pathogens [9,10,11,12,13,14]. For instance, *E. albertii* commonly carries the *eae* gene that encodes intimin, an important virulence factor also harbored by pathogenic subgroups of *E. coli* [10,13,15,16]. This has probably resulted in underestimation of *E. albertii* infection numbers. For example, in several gastroenteritis outbreaks the causative pathogen was wrongly diagnosed as EPEC instead of *eae*-positive *E. albertii* [9,11,12,14,17,18]. Because of this, several enrichment broths, isolation, and identification protocols have been developed, however, to date, there is no standard method [19,20,21,22,23]. The specificity and sensitivity for most of these protocols though high [24], have been evaluated based on a limited number of strains, thus it is not clear if such protocols cover the biodiversity of *E. albertii* isolates. In addition, the suggested further identification and characterization approaches such as *E. albertii*-specific PCR, multilocus sequence type (MLST) analysis, and O-genotyping are too costly, time-consuming, or not available in most routine laboratories, especially in resource-limited areas where such cases might occur more frequently [4,25,26]. Coupled with the fact that *E. albertii* is underappreciated as an enteric pathogen [27], it reduces the chance of it being included on the differential diagnosis list, further compounding identification. Therefore, *E. albertii* misclassification will probably continue until a novel, simple and accurate diagnostic tool is created. Moreover, awareness of this enteropathogenic *Escherichia* species needs to be increased.

*E. albertii* infections mostly present as watery diarrhea, fever, and abdominal pain, with most cases resolving without complications [5,14]. Genes such as *eae*, associated with attaching and effacing lesions, *cdt* cytolethal distending, and Shiga toxins (*stx*), among others, make part of the virulence arsenal contributing to this pathogens’ clinical manifestations [5,11]. Early outbreak and sporadic case reports indicate that *E. albertii* is transmitted through contaminated water and food such as salads, chicken, and packed lunches [9,11,12,14,28]. Due to poor hygiene and sanitary conditions that prevail in several countries, such infectious gastroenteritis will remain an important cause of morbidity and mortality [29,30]. Therefore, we need to expand our knowledge on infectious gastroenteritis causative agents to better control and prevent such infections. In this article, we review recent advancements in our understanding of key aspects, e.g., the biochemistry and epidemiology of *E. albertii* (Figure 1). Emphasis is made on the currently available isolation and identification diagnostic modalities. Knowledge gaps are highlighted to provide a basis for future research on this emerging elusive zoonotic pathogen.

## 2. Biochemical Properties of *Escherichia albertii*

Initially, the biochemical properties of *E. albertii* were clear with key biochemical makers being an inability to produce indole from tryptophan or to ferment lactose, xylose and rhamnose [1,10,20,31,32,33]. These phenotypes are due to the absence of effector proteins such as those encoded by the *rhaMDABSRT* (rhamnose utilization) and *xylBAFGHR* (xylose utilization) operons required for these metabolic functions in *E. albertii* [13]. Based on a limited number of strains, *E. albertii* was also initially reported as unable to ferment sucrose and melibiose [1,32]. However, as the number of available isolates increased this narrative changed. Some *E. albertii* strains have been confirmed to be lactose, melibiose, and sucrose fermenters as well as capable of producing indole from tryptophan [26,34,35,36,37]. Interestingly, some of these strains lack the known genes required for the utilization of these sugars, suggestive that they may encode yet-to-be-described pathways for utilization of these carbon sources. For instance, *E. albertii* strain NIAH_Bird_23 showed a weak lactose fermentation phenotype but lacks the *lacA*, *lacY*, and *lacI* genes required for lactose utilization [13].

With further analysis of more isolates, additional phenotypic variability among *E. albertii* strains is being observed including utilization of maltose, d-mannitol, d-sorbitol, trehalose, and ortho-nitrophenyl-B-galactoside (ONPG), esculin hydrolysis, β-galactosidase, lysine-decarboxylase and l-prolineaminopeptidase activity (Table 1) [10,33,38]. It is likely that as more isolates are characterized, variability in these biochemical properties will also increase, giving a clearer picture of the true *E. albertii* distinguishing features. It has been postulated that such phenotypic variability is suggestive that biotypes or biovars may exist within *E. albertii*. Using some of these phenotypes, three biogroups have been described (Table 2) [36], however, not all *E. albertii* strains fit in these groups [39].

According to currently available data, isolates of this pathogen analyzed so far, can reduce nitrate, assimilate 3-hydroxybenzoate, utilize glucose, D-mannose, mannitol, and galactose, and are positive for methyl red (Table 1) [1,27,32,38]. This bacterium does not grow on KCN medium or produce H_2_S and is nonmotile, however, *E. albertii* carries fully conserved flagellar biosynthesis genes [2,13,20,33]. Most *E. albertii* strains do not utilize several common and uncommon sugars, such as adonitol, cellobiose, 2-ketogluconate, myoinositol, rhamnose, melibiose xylose, and xylitol, phenotypes potentially exploitable for *E. albertii* differentiation from other *Enterobacteriaceae* [26,32,35,37,38]. However, the feasibility of using some of these sugars in routine diagnostic media is restricted due to their cost. *E. albertii* is also negative for chitinase, oxidase, hydroxyproline deaminase, tripeptidase, and proline deaminase (Table 1) [1,10,31,32,33]. Although not commonly used in most routine laboratories, such phenotypes might be useful in distinguishing *E. albertii* from other species such as *H. alvei* and *E. coli*.

## 3. Detection and Identification Methods for *Escherichia albertii*

Discriminating *E. albertii* from other *Enterobacteriaceae* is challenging. This zoonotic pathogen is often misclassified as *E. coli*, *H. alvei*, *Shigella boydi* serotype 13, *Salmonella enterica*, and *Yersinia ruckeri* due to their similar phenotypic and genetic features [2,7,10,19,26,31,32,41,43]. Moreover, no commercial test is available to differentiate *E. albertii* from other *Enterobacteriaceae*. On the systems that currently exist, *E. albertii* isolates are often misidentified. For instance, on the MicroScan dried overnight panels and Micronaut-E system, they are identified as *Y. ruckeri* while on the API 20E and Vitek systems some strains are identified as *H. alvei* [32,36,38].

A multitude of different protocols have been proposed for differentiating *E. albertii* from other *Enterobacteriaceae*, however, since these methods have been validated using a minimum number of strains from limited sources, the extent of their accuracy is questionable. Moreover, these methods especially the agar and enrichment broths are not selective for *E. albertii*, most of them are modifications of methods used for *Enterobacteriaceae*, hence other species within this family can grow [23]. Moreover, earlier protocols were based on the now invalidated hypothesis that no *E. albertii* strains can ferment sucrose and lactose which might have also led to the misclassification of some *E. albertii* strains [2,9,10,12,21,33,34,44,45,46,47].

### 3.1. Enrichment Broths

Most *E. albertii* isolation protocols rely on a single enrichment step using various broths including buffered peptone water (BPW), *E. coli* broth (EC), modified EC (mEC), novobiocin–mEC (NmEC), tryptic soy broth (TSB), modified TSB (mTSB), or novobiocin–cefixime–tellurite supplemented mTSB (NCT-mTSB) at various temperatures ranging from 20 to 44.5 °C [22,23,37,46,48,49,50]. The added novobiocin (20 to 25 mg/L) inhibits most Gram-positive bacteria, while in the NCT-mTSM enrichment broth the tellurite (1 mg/L) and cerufuxim (0.05 mg/L) inhibit most Enterobacteria, while growth at temperatures ≥ 42 °C is postulated to inhibit most background microbiota, *Shigella* spp. and *E. coli* [22,23,37]. Enrichment protocols that employ antimicrobial selective pressure though promising, run the risk of selecting only *E. albertii* strains resistant to these antimicrobial agents especially when they are validated using a few strains from limited sources and or geographical regions. Since most *E. albertii* were initially identified as *E. coli*, one can speculate that *Enterobacteriaceae* enrichment broth (EB) is a good option for the enrichment of samples for *E. albertii* isolation. However, since some *Enterobacteriaceae* selective media use lactose as the main carbon source, caution must be exercised as most *E. albertii* do not ferment lactose and might be outcompeted by lactose fermenting *Enterobacteriaceae* [10,36].

### 3.2. Molecular Approaches

Several PCR protocols have been proposed, most of which are based on the *lysP*, *mdh*, *clpX*, *yejH*, and *yejK*, *EAKF1*_ch4033, and the species-specific *cdt* gene [13,19,22,24,33,51]. The *eae* gene was also proposed as a gene marker for *E. albertii* but this was soon discarded as several pathogenic *E. coli* strains also carry this gene [5,24]. This approach also discriminates against *eae*-negative *E. albertii* isolates [6]. In addition, some studies have reported nonspecific amplification in assays targeting *lysP* and *mdh* genes indicative of lower sensitivity for such protocols [24,52]. Using 67 *E. albertii* strains Hinenoya et al. [24], observed varying sensitivities from 98.5 to 100% of different PCR assays developed by Ooka et al. [13], Hyma et al. [19], and Lindsey et al. [51].

Currently, it seems the DNA-binding transcriptional activator of the cysteine biosynthesis gene (*EAKF1_*ch4033) and the *cdtB* gene-based PCR assays are receiving the most attention. Three potential *E. albertii* specific genes (*EAKF1*_ch3804, *EAKF1*_ch4075c and *EAKF1*_ch0408c) have also been proposed [6].

On the National Center for Biotechnology Information database (NCBI; https://www.ncbi.nlm.nih.gov/; accessed on 18 March 2022), there are currently 319 *E. albertii* whole genome sequences (WGS) available. Using an in-silico PCR analysis approach, we observed that the *E. albertii cdt* gene primers developed by Hinenoya et al. [24] correctly identify 310 strains (96.1%) as *E. albertii* and three possibly misclassified strains as non-*E. albertii* (Table 3). Previous studies have shown 100% specificity and sensitivity of this PCR assay, although a limited number of strains (*n* = 64) was analyzed [24]. The *E. albertii* cytolethal distending toxin (*Eacdt*) gene is homologous to the *E. coli cdt-II* gene, which is one of five variants of this gene in *E. coli* (*cdt-I* to *cdt-V*) [53], suggesting that *E. coli* strains encoding this *cdt-II* gene are misclassified *E. albertii* [26,41,43].

The *E. albertii* specific primers proposed by Lindsey et al. [51] targeting *EAKF1*_ch4033, correctly identified 305 (96.5%) strains and also distinguished the aforementioned three misclassified strains (Table 3). Our analysis shows that these three misclassified strains might belong to the novel species *Escherichia ruysiae* [3]. Since *E. albertii* and *E. ruysiae* have similar biochemistry phenotypes, misidentification of the two species can occur via such tests as well. The three presumably misclassified *E. ruysiae* strains are isolates 385522, 832738, and 896071, from whole-genome shotgun sequencing projects GenBank: AAVTNX000000000.1, AAWWUJ000000000.1 and AAWWXV000000000.1, respectively. The strains for which the in-silico PCR did not work either lost or had different alleles of these genes or the sequencing quality was not good enough. An *E. albertii* strain isolated from an asymptomatic cat lacked the *Eacdt* gene due to a deletion mutation [13]. Luo et al. [6], also reported four isolates that lacked the *EAKF1*_ch4033 gene. Because of these reasons, some strains which are true *E. albertii* will be missed by this approach, however, combining the two primers sets in a multiplex PCR would correctly identify all 316 true *E. albertii* isolates.

Using the in-silico PCR approach described above on the currently available 178 *E. coli* WGS with inconclusive taxonomy on NCBI, revealed approximately 15% (*n* = 26) misclassified *E. albertii* strains positive for the *Eacdt* and or *EAKF1*_ch4033 genes (Table 3). These 26 genomes were further confirmed to be *E. albertii* via MLST analysis. This shows that even with WGS, some strains can still be misclassified, an indication that genome storage databases need to update their taxonomy assigning settings. Interestingly, a previous study showed that 26 of presumed 179 (14.5%) *eae*-positive EPEC or EHEC isolates were misclassified *E. albertii* [10]. In another study that analyzed 373 *eae*-positive *E. coli* strains using biochemical tests, MLST, and an *E. albertii*-specific PCR, 17 (4.6%) isolates were reidentified as *E. albertii* [26]. 

Sequencing of 16S rRNA and or *rpoB* genes has been put forward as a discriminatory tool for *E. albertii* identification [1,2,19,21]. However, studies have demonstrated that, although, *rpoB* sequencing is promising [21], the differences between 16S rRNA genes of *Shigella* spp., *E. coli*, *E. fergusonii*, and *E. albertii* are insufficient to effectively distinguish them [54,55]. Thus, 16S sequencing is not the best system for differentiating these pathogens.

### 3.3. Selective Agar Plates 

Post enrichment, cultures are plated on deoxycholate hydrogen sulphide lactose (DHL) or MacConkey (MC) agar and various variations of these media with 1% xylose, 1% rhamnose, and or 1% melibiose supplementation (XR-MC, XR-DHL, XRM-MC, XRM-DHL, and XR-DH) [22,36,56]. *E. coli* utilize these sugars and grow to produce colored colonies. On the other hand, most *E. albertii* isolates described to date cannot ferment the above-mentioned sugars and most do not ferment sucrose and lactose, hence white colonies on such plates are presumed to be *E. albertii* often followed by further validation through *E. albertii* specific PCR. Some protocols recommend to only plate *E. albertii*-PCR-positive enriched samples [22]. From our analyses, using the *Eacdt* and *EAKF1*_ch4033 gene primers in a multiplex PCR assay would ensure 100% identification. Other genes such as *lysP*, *mdh,* and *clpX* have been used at this step to differentiate *E. albertii* from other *Enterobacteriaceae* through multiplex PCR assays, however, they do not detect all *E. albertii* and can result in false positives [19,20,21,52]. 

Melibiose, sucrose, and lactose fermenting *E. albertii* strains have been reported meaning that the selection of white colonies only might result in the exclusion of other *E. albertii* isolates. In support of this, several studies have reported *E. albertii* PCR positive samples from which the bacterium could not be isolated [22,37,46,49]. For instance, Hinenoya, et al., [49] reported a recovery rate of 25% (*n* = 62) from 248 PCR positive raccoon rectal swab samples. Although there might be other explanations for these observations, it is likely that the selection of white colonies (strains that do not ferment lactose, sucrose, or melibiose) might have resulted in the non-inclusion of colored *E. albertii* colonies from lactose or melibiose fermenting isolates when such approaches were employed. The XRM-MC agar [56] and the recently proposed XR-DH agar [23], lack lactose or both sucrose and lactose, respectively, circumventing the lactose or sucrose fermentation issue, however, other non-*E. albertii* strains such as *Shigella* spp. also produce white colonies on these plates introducing another selection challenge. 

The chromogenic mEA (*E. albertii* medium) agar, proposed by Maheux et al. [35], contains cellobiose and peptones as carbon sources also circumventing the lactose and sucrose fermentation challenges. This agar allows isolation of both lactose-positive and -negative *E. albertii* strains. Bile salts and incubation at 44.5 °C inhibits the growth of most background non-enteric bacteria. β-D-glucuronidase-positive bacteria grow as blue colonies while bacteria that can ferment cellobiose produce acid and appear as pink colonies. Cellobiose non-fermenting bacteria, including *E. albertii* produce ammonia from peptone metabolism, thus growing as white/colorless colonies. However, at least 18 other species including *H. alvei*, *Acinetobacter baumannii*, *Bordetella* spp., *Citrobacter* spp., *E. fergusonii*, *E. hermannii*, *Proteus mirabilis*, *Salmonella bongori*, *S. enterica*, and some *Shigella* spp., give false-positive results as they also grow as white colonies on mEA agar [35]. This limits the applicability and selectiveness of this agar especially in samples containing *E. albertii* together with such species.

Interestingly, a few *E. albertii* strains have been reported to also utilize xylose (<9%) and rhamnose (<5%) [2,10,31,32,33]. Since inability to utilize these two carbon sources is key in most isolation and identification protocols, caution must be taken when selecting colonies to test especially from PCR-positive cultures with no white colonies. 

#### Matrix-Assisted Laser Desorption Ionization-Time of Flight Mass Spectrometry

Matrix-assisted laser desorption ionization-time of flight mass spectrometry (MALDI-TOF-MS) is an accurate, cost-effective, fast, high-throughput method used for the identification and typing of microorganisms [57,58,59]. Recently, Hatanaka et al. [60] developed a MALDI-TOF-MS-based *E. albertii* specific identification protocol and mass spectra library which can be used to accurately distinguish *E. albertii* from *E. coli*. Using their own refined *E. albertii* database library and species-specific spectral peaks the authors could correctly identify all tested (*n* = 58) *E. albertii* strains [60]. Although, MALDI-TOF-MS is a promising tool for differentiating *E. albertii* from other *Enterobacteriaceae*, its use is restricted due to high initial investment cost and limited availability in most laboratories. Moreover, since only the mass spectra for *E. coli* identification are established, *E. albertii*-specific database libraries need to be created locally to be able to effectively distinguish this emerging pathogen from other species. For instance, in the above study, when using the manufacturers’ database library only 4 of the 58 tested strains (<7%) were correctly identified [60]. 

Another recent study reported that with custom-made or commercially available databases, MALDI-TOF-MS was not able to accurately differentiate a significant number of *E. coli* and *Shigella* spp. isolates [61]. An earlier study by others, also using a custom reference library, correctly classified 94.4% of the 180 tested strains as *E. coli* or *Shigella* spp., but incorrectly classified six (3.3%) isolates, with the results of four (2.2%) strains being non-interpretable [62]. These observations plus the high genetic similarity of *E. coli*, *Shigella* spp., and *E. albertii* indicate that any new *E. albertii* database must similarly be validated for specificity against *Shigella* spp. since *E. albertii* is also often misclassified as *Shigella boydii*. From our own observation, the novel species *E. ruysiae* can be misclassified as *E. albertii*, hence it would be prudent to validate any created database against this species as well. Overall, when coupled with its potential to subtype strains and detect antimicrobial resistances determinates/profiles, MALDI-TOF-MS applied in combination with an *E. albertii* specific database library can be a superior option over PCR for differentiating *E. albertii* [58,59,60].

### 3.4. Characterization by MLST and O-Genotyping

Multilocus sequence analysis of seven housekeeping genes (*adk*, *fumC*, *gyrB*, *icd*, *mdh*, *purA*, and *recA*), has been proposed as a more accurate discriminatory tool to differentiate between *E. albertii* and *E. coli*, however, such a diagnostic tool is not always available in routine diagnostic labs [2,10,13,19,25]. Its use is further hindered by the cost and time-consuming nature of such an approach, making it less favored in most routine labs. Since there is no *E. albertii* specific serotyping system yet, further strain characterization can be performed by O-genotyping which can assist in understanding *E. albertii* strain diversity [4]. This system examines the genetic structure of O-antigen gene clusters of this pathogen. To date, forty of such O-genotypes (EAOg1 to EAOg40) have been defined [4,47,63]. These genotypes are detectable via a multiplex PCR-based system targeting the *wzx* genes and an *E. albertii*-specific gene [4,5]. This is because the *wzx* sequences of several EAOgs show high similarity (>95%) to those of other species including *E. coli* and *Shigella* serotypes [5,64,65]. However, according to Gomes et al. [5], this system could only genotype around 82% of the analyzed *E. albertii* genomes, indicating that the system needs to be further improved to cover the remaining 18% strains. 

Nakae et al. [66], developed an H-genotyping system to complement the O-genotypes. This system is based on the *fliC* gene which encodes flagellin. To date, four *E. albertii* H-genotypes (EAHg1–EAHg4), which are distinct from the 53 known *E. coli* H-antigens, have been described. Similar to O-genotyping, a multiplex PCR-based H-genotyping system has been developed [66]. However, like most *E. albertii* identification and classification systems, analysis of more *E. albertii* strains is required to further validate this typing system.

### 3.5. Escherichia albertii Whole Genome Sequencing

Because *E. albertii* is challenging to classify using traditional methods, WGS is the gold standard for identification and further characterization of this pathogen and must always be considered especially in outbreak situations. Besides aiding in outbreak source tracking, it has the added advantage of giving more details on virulence potential, antimicrobial resistance profiles, and stress tolerance capabilities, due to the detection of genes linked to these traits [4,5,6,13,67]. However, WGS application is hampered by its cost and limited availability in some laboratories. As observed with the misclassified *E. coli* and *E. ruysiae* genomes described above, WGS is not immune to error. This misclassification is most likely due to the high genomic similarity of *E. albertii* and other *Escherichia* species [5,6,13]. On average, the chromosome of *E. albertii* is analogous in size to that of *E. fergusonii* but is smaller than that of *E. coli* [5,13]. Average nucleotide identities (ANIs) ranging between 86 to 90% have been observed between *E. albertii* and other *Escherichia* species [5,13]. Some of these differences and their phenotypic consequences such as the inability to utilize rhamnose have been highlighted in this review. Complete 16S rRNA gene sequencing and DNA-DNA hybridization studies have shown a significant difference between *E. albertii* and its initial misclassification identity *H. alvei*, with relatedness values of 93.5% and 9 to 17%, respectively [1]. Interestingly, *E. albertii* strains are highly conserved with ANI values > 98%, with no major phylogenetic relationship differences between animal and human isolates [5,6,13,67]. For instance, Wang et al. [67] recently reported 4 chicken *E. albertii* strains from the United States of America, which are evolutionarily very close to a human strain isolated from a case of acute diarrhea in Bangladesh. 

*E. albertii* strains can thus far be grouped into two clades (clades 1 and 2) that divide into five distinct phylogroups (G1 to G5) and eight lineages, most of which belong to clade 2 (lineage 2 to 8) [4,5,6]. Due to such high genomic similarity among *E. albertii* strains, the development of *E. albertii*-specific primers that detect most, not if all isolates, is possible. However, since the currently available genome-based studies used a limited number of strains from limited sources, studies analyzing more genomes from different sources and geographic locations are required. For more in-depth genome comparison please see the studies by Luo et al. [6] and Ooka et al. [13]. A summary of the *E. albertii* isolation and identification protocols is given in Figure 2.

## 4. Occurrence of *Escherichia albertii* in Animals

*E. albertii* has been isolated from various domestic and migratory birds, occasionally causing epidemics worldwide [2,50,67,68,69,70,71]. *E. albertii* has also been reported in pigs, cats, dogs, bats, raccoons, penguins, and seals [10,41,49,70,72]. Isolation frequency seems highest in birds [44,46,48], hence poultry could act as a reservoir and its meat might pose an increased risk for *E. albertii* exposure and or infection [50,67]. For instance, surveys in Australia between 1994–2004 and 2010–2011 showed the prevalence of *E. albertii* of 14.3 to 18% in magpies and 6.7 to 33% in chickens [70,73]. Interestingly, the same analysis did not detect *E. albertii* in fish, snakes, lizards, crocodiles, and frogs, over a 10-year period [70]. Isolation of this pathogen in raccoons (57.7% prevalence; [49]), bats, seals, and wild birds is suggestive that wild animals more so birds might be reservoirs of this bacteria. It would be advantageous to evaluate *E. albertii* prevalence in different wild animals and other livestock such as cattle, goats, and sheep from which this pathogen has not yet been detected. However, the isolation of *E. albertii* from mutton meat supports the occurrence of this pathogen in livestock [48]. The occurrence of *E. albertii* in apparently healthy animals might suggest the existence of careers that can act as reservoirs of this pathogen in livestock [2,41,71]. *E. albertii* in companion animals such as cats and dogs as well as farm animals such as chickens and pigs might act as a source or reservoir of this pathogen for people in close contact with these animals. Moreover, animal and human *E. albertii* isolates are heterogeneous and seem not to be host-specific, therefore, all its strains might have zoonotic potential and represent a significant public health threat [6,50,67].

The clinical picture of active *E. albertii* infections in animals is rarely described. This might be a consequence of infected animals being subclinical or dying acutely without many observable clinical signs. This possibility coupled with the challenge in differentiating this pathogen from *E. coli* might result in illness due to *E. albertii* infection being misdiagnosed, precluding full definition of its clinical presentation both in humans and animals.

## 5. Occurrence of *Escherichia albertii* in Food

*E. albertii* has been isolated from the environment, water, packed lunch, lettuce, salad, pork, chicken, giblets, mutton, duck meat, minced meat, and Damietta cheese indicating that this bacterium can be transmitted through food and water [10,12,17,21,23,25,44,46,48,74,75]. In most outbreaks and sporadic cases, the contaminated vehicle for disease transmission is not identified. Nonetheless, in a few cases, it has been confirmed that salads, water, and chicken are some of the contaminated vehicles [12,14,18,70]. *E. albertii* was associated with outbreaks in campers who drank contaminated water [14,17]. Furthermore, it has been detected in various water bodies across the world [25,27,39,75]. The occurrence of *E. albertii* in water is suggestive that seafood is also potentially contaminated with this pathogen. The study by Arai et al. [22] confirmed this as they detected *E. albertii* in 1.9% of 427 samples of raw Pacific and Japanese rock oysters. The occurrence of this pathogen in other kinds of seafood need to be elucidated. The use of contaminated irrigation water might be responsible for contaminating plants such as lettuce and eventually salads, hence caution must be exercised when using untreated irrigation water, especially in areas where *E. albertii* contaminated water is expected. 

Since the full phenotypic features such as growth potential on different foods and hurdle technique resistance profiles of *E. albertii* are not yet known, it is challenging to determine the most effective approaches to control this pathogen on food. It is tempting to speculate that hurdle procedures employed for *E. coli* and other *Enterobacteriaceae* coupled with good manufacturing practice can minimize *E. albertii* occurrence in foods. Interestingly, Sharma et al. [76] using five strains reported that *E. albertii* is less tolerant to heat, acid, and pressure than *E. coli* O157:H7. Moreover, it is highly likely that *E. albertii* is more sensitive to osmotic stress compared to *E. coli* due to the absence of the *betIAB* operon required for glycine betaine synthesis an important osmo-protective solute [13,77]. However, this needs to be experimentally confirmed. The potential transmission cycle of *E. albertii* is depicted in Figure 3.

## 6. Pathogenesis and Virulence Factors of *Escherichia albertii*

*E. albertii* is one of the attaching and effacing pathogens, a phenotype confirmed in vivo using the rabbit ileal loop model [8]. Its pathogenesis depends on its ability to adhere to epithelial cells with the formation of attaching-effacing lesions. *E. albertii* achieves this through the dual activity of the type III secretion system effectors and an outer membrane protein (intimin), encoded by a locus of enterocyte effacement (LEE) and the *eae* gene, respectively [5,13,78]. Similar effacing lesions are produced by other pathogens such as EHEC and EPEC as well as *Citrobacter rodentium* [5]. These lesions promote the bacterial invasion process, while the ability of *E. albertii* to survive intracellularly protects it from intestinal clearance and the immune system, causing prolongation of the diarrheal disease [5]. Invasion of epithelial cells was shown to depend on intimin interactions with its receptor Tir, located on the host cells membrane [79]. This intimin-Tir interaction results in actin polymerization and the formation of pedestal-like structures beneath adherent bacteria [80,81,82,83,84]. It has also been demonstrated that *E. albertii* can invade HeLa, Caco-2, and T84 cells, and can decrease transepithelial electrical resistance by redistributing tight junction proteins (zona occludenes-1) which leads to higher cell permeability [78,85,86,87].

Interestingly, in *E. albertii* the highly conserved LEE is integrated almost exclusively on the *pheU* tRNA locus while in EHEC or EPEC, LEE is found either at the *pheU*, *pheV*, or *seIC* loci suggestive of multiple acquisition events in *E. coli* [5,10,13,26,27,88,89]. In further contrast to *E. coli*, some *E. albertii* strains contain an intact *E. coli* type III secretion system 2 (ETT2) locus which includes the virulence genes *yqeH*, *ygeF*, *eprH*, *epaS*, *eivG*, and *etrA*, and is integrated at the *glyU* tRNA locus, however, the importance of this locus for virulence is not yet clear [13,14]. In addition, porcine attaching-effacing associated protein encoded by *paa* is highly conserved in *E. albertii* strains, but its occurrence is strain-specific [90]. 

Almost all reported *E. albertii* isolates carry a *cdtABC* locus which encodes the cytolethal distending toxin (Cdt). Of the 5-subtype known to exist in *E. coli* [53], Cdt-I and Cdt-II have been reported in *E. albertii*, with Cdt-II being the predominate subtype [6,91]. Only a single strain that encodes both Cdt-I and Cdt-II subtypes has been described to date [91]. Moreover, a new subtype, Cdt-VI was recently reported in *E. albertii* [6]. The highly conserved *EacdtABC* operon has so far mostly been detected on the chromosome and not in mobile genetic elements [13,26,43]. However, Gomes et al. [5] recently reported two strains that carry the *cdt* gene in a prophage region. Cdt protein is composed of three subunits, CdtA and CdtC are critical for translocating the virulence factor CdtB into the host cell. CdtB has DNase I activity and functions through DNA damage, inducing G2/M cell cycle arrest and eventual apoptosis [53,92,93,94]. Although the importance and role played by Cdt in *E. albertii* pathogenesis is not yet fully understood, it has been demonstrated that Cdt is linked with persistent colonization and invasion by bacteria, which, in turn, affects disease severity [95,96]. Since *cdtB* is highly conserved amongst *E. albertii* strains it is now being exploited as a genetic marker for the identification of this pathogen.

Some *E. albertii* encode the protein synthesis inhibiting cytotoxic Shiga toxin 2 (Stx2a or Stx2f) henceforth referred to as Shiga toxin-producing *E. albertii* (STEA) [5,10,13,26,33,39,45,49,67,91,97,98,99]. Due to this carriage of the *stx* gene, some *E. albertii* isolates have been misclassified as EHEC [10,45]. The Stx2f variant is more common among *E. albertii* strains [45], with only a few strains encoding the Stx2a subtype being reported to date [5,45]. It seems carriage of the Stx2a toxin increases virulence with bloody diarrhea being one of the consequences of infection [45,100]. 

Using the CHO and Vero cell cytotoxic assays, studies have shown that some *E. albertii* strains produce biologically active Cdt and Stx toxins, respectively [10,20,26,49,91,99]. As seen with Shiga toxin-producing *E. coli* (STEC), it is possible that STEA can cause hemorrhagic colitis, hemolytic-uremic syndrome (HUS), thrombotic microangiopathy, and renal failure [13]. As observed in *E. coli*, such clinical manifestation is highly likely in strains expressing the *stx*2a allele [101,102,103]. In *E. albertii*, the Shiga toxin is mainly harbored on phages, hence clinicians must be aware of the danger of using antibiotics that might induce *stx*2-harboring bacteriophages, which might worsen the prognosis in STEA infections [104]. In support, some studies have shown that STEA culture supernatants’ toxicity is enhanced by mitomycin C, implying that the *stx* genes in these isolates, might be harbored on inducible prophages [49].

It is not clear if pathogenicity differences exist between *E. albertii* strains. However, the presence, distribution, and variants of virulence genes in its strains vary with lineage, suggestive of potential virulence variation [6]. Hinenoya et al. [50], using chicken colonization experiments showed that human isolates have a higher colonization rate than wild bird-derived isolates. Over 44 potential virulence factors have been identified in *E. albertii*, including those encoded by, *ibeA*, *sepL*, *ent* and *fep*, *iuc-ABCD*, *iutA*, *ehxA*, *entABCES*, *fepABCDG*, *hlyABCD*, *map*, *espA*, *espB*, *nleA*, and *phoE* [6,13,14,67]. These virulence factors contribute towards invasion, virulence factor translocation and regulation, iron acquisition and transport, cytolysis, immune response evasion and dampening, and stress tolerance [33,48,105,106,107,108,109]. This list of virulence factors will continue to grow as more sequence data become available. Therefore, much work is required to elucidate the regulation, structure, and mechanism of action of these virulence factors and their role in the pathogenicity of this bacterium. Table 4 summarizes some of the major virulence factors detected in *E. albertii.* For a more detailed review of *E. albertii* pathogenesis please refer to Gomes et al. [5].

## 7. Clinical Significance of *Escherichia albertii*

Early evidence suggests that *E. albertii* may be present in animals or humans as a commensal or pathogen. The clinical significance of *E. albertii* is not yet fully understood, partially because it is difficult to discriminate it from other *Enterobacteriaceae* using routine identification protocols. Nevertheless, infectious diarrhea outbreaks and sporadic cases due to *E. albertii* have been reported in multiple countries including Antarctica, Alaska, Bangladesh, Belgium, Brazil, China, Germany, Guinea-Bissau, Iran, Japan, Mexico, Nigeria, Poland, and the United States of America, signifying a worldwide distribution [2,7,9,11,12,14,48,50,67,72,90,99,110,111,112]. Despite this and several retrospective studies determining that many EPEC and EHEC isolates implicated in disease outbreaks are misidentified *E. albertii* [9,10,11,12,14,17,18,44], this pathogen is still often overlooked as a causative agent for infectious diarrhea [27].

*E. albertii* infections mainly present as acute watery diarrhea, fever, and abdominal pain with occasional reports of headache, nausea, dehydration, and abdominal distension [1,7,10,14,112]. In most cases, these infections are self-limiting, with patients often recovering with little to no treatment [14]. This could be suggestive of reduced virulence in *E. albertii* compared to *Shigella* spp. and enteroinvasive *E. coli*. In support of this hypothesis, *E. albertii* lacks *invE* or *ipaH*, which encode important virulence factors in *Shigella* and enteroinvasive *E. coli* [13].

Afshin et al. [113] and Zaki et al. [114] reported five and seven cases of *E. albertii*-associated urinary tract infections, respectively. A single case of *E. albertii* bacteremia has been reported in an old lady with multiple comorbidities probably because of its ability to translocate from the intestinal lumen to other extraintestinal sites [5,34,115]. This might be a rare clinical presentation, however, due to the potential for misclassification, *E. albertii* bacteremia might occur more frequently but is misclassified as the more common *E. coli*. It would be prudent in the future to consider *E. albertii* in *Enterobacteriaceae* bacteremia presumptively due to *E. coli* especially those that have discrepant results for routine etiology molecular identification. In support of this, retrospective analyses have indicated that *E. albertii* represents a significant portion of strains identified as *eae*-positive *E. coli* including Shiga toxin 2f–producing strains [12,17]. Because *E. albertii* possesses the *eae* gene, many strains might have been misidentified as EHEC or EPEC. 

Incubation periods of *E. albertii* infections which present as diarrhea are relatively short (average 12–24 h) while mortality and morbidity rates are unknown [14]. To date, several outbreaks have been reported mainly in Japan, with most of these initially misidentified as *E. coli* outbreaks [10,12,14,17]. In all these outbreaks and other sporadic cases, it appears mortality rates were low or none, however, morbidity rates appeared to be relatively high, i.e., >50% of the exposed population [14]. For instance, in two outbreaks in Japan, one linked to boxed lunches and another associated with eating at the same restaurant, 20 of 31 (64.5%) and 48 of 94 (51.1%) exposed people became ill, respectively [11,12,14]. While one of the largest recorded *E. albertii* outbreaks, linked to contaminated water had an attack rate of 66.7% (273) among the 409 potentially exposed individuals [14].

Because *E. albertii* and *E. coli* share ecological niches and display high genomic similarity such as virulence and stress tolerance effectors [5,6,13,27], one can speculate that these pathogens might pose similar public health threats, therefore measures employed to mitigate EPEC and EHEC might also be required for this emerging zoonotic pathogen. 

As with most bacteria, antimicrobial resistance is a major concern. Several strains of *E. albertii* have been shown to be resistant to a significant number of important antibiotics including tetracycline, macrolides except for azithromycin, lincosamides, ampicillin, penicillin G, oxacillin, gentamicin, ciprofloxacin, trimethoprim, sulfamethoxazole, fusidic acid, rifampicin, meropenenem, imipenem, and norfloxacin [26,38,50,51,67,90,114]. Moreover, multidrug-resistant *E. albertii* strains [67], that displayed resistance or encode resistance determinates to antibiotics from at least eleven classes have been reported [6,116]. Combined with the virulence potential of this pathogen such strains pose a significant public health threat. Therefore, to effectively prevent and control its infections, much must be done to fully understand all aspects of this emerging zoonotic pathogen including its antimicrobial resistance determinants. 

## 8. Risk Assessment

The infectious dose, transmission routes, incidence rates, prevalence, epidemiology, and predisposing factors of *E. albertii* have not yet been fully elucidated, making risk assessment difficult. However, early indications suggest that children below 10 years and immunosuppressed individuals with multiple commodities might be at higher risk, but outbreaks have also been reported in seemly healthy people [5,7,8,12,14,34,45,113,114]. Poor hygiene conditions especially during food preparation, consumption of raw or minimally cooked meat particularly poultry, and drinking untreated water seems to increase the likelihood of *E. albertii* infection [14,17,18,112]. The increased occurrence of clinical disease in younger age groups might be a true association or a reflection of sampling bias. Enteric infections may have more severe negative health outcomes amongst young children [117], such as physical and cognitive development impairments [118], hence in most studies, this population is preferentially targeted. Moreover, in STEC infections, HUS development is more likely in children <5 years old [119]. Though not confirmed we speculate that the risk of HUS development in STEA infection might be higher in children. Dual carriage of *stx*2 and *eae* genes reported in some *E. albertii* strains might increase the risk of HUS development [5,6,45]. While *E. albertii* strains that lack this *eae* gene, such as those isolated from water and chicken carcass rinse water might be less pathogenic and pose a lower risk for severe clinical disease [21,25,75].

Some *E. albertii* strains have been observed to produce biofilms [90,120], a phenotype known to increase resistance against different stresses and the host immune system including the mechanical movements of intestinal peristalsis [121,122]. This might increase the severity and duration of the disease, complicating treatment outcomes. Moreover, biofilm formation can assist in niche colonization and persistence in food processing plants, however, further studies are required to confirm this.

## 9. Knowledge Gaps and Future Research Needs

*E. albertii* is an emerging zoonotic foodborne enteropathogenic pathogen, however, there are still many knowledge gaps that need to be bridged to fully understand and be able to effectively control and mitigate the risks this pathogen poses. We need to understand fully its pathogenic potential and clinical relevance. Critical information such as mortality and morbidity rates, infectious dose, and predisposing factors need to be elucidated. Its full epidemiological picture, transmission routes, reservoirs both in water bodies, humans, livestock, and wildlife need to be further elucidated. Risk assessment including characterization of *E. albertii* strains growth capacity on different foods and hurdle technique resistance profiles are required. 

Due to challenges in isolation and identification of this pathogen, more specific enrichment, isolation, and identification tools are required. Protocols that employ PCR-based methods have been developed using a limited number of strains, from limited sources and or regions, hence these primers might be selective for these strains missing *E. albertii* strains with other alleles of the target genes. Such protocols need to be reviewed using more strains, primers must be validated using more genomes from a diverse strain collection. The three presumed *E. ruysiae* strains potentially misclassified as *E. albertii* we identified show that much work still needs to be done to correctly classify this species even at the genome level. It is highly likely that with the discovery of new novel species misclassification will continue to be a problem. Sequence databases need to continually filter and update their genome records to ensure that they are in line with the new identification systems. 

The discovery of lactose fermenting *E. albertii* isolates further complicates isolation and identification, therefore, a selective plate that can effectively identify both strains able and not able to ferment lactose is urgently needed. Most previous protocols selected for non-lactose fermenting *E. albertii* [22,37,112], it will be interesting to investigate how many *E. albertii* strains were missed using such an approach. Reports of antimicrobial resistance in *E. albertii* especially toward extended-spectrum beta-lactam antibiotics are troubling. The extent of such antimicrobial resistance in this species needs further investigation. The clinical and economic importance of *E. albertii* is currently unknown. Until routine laboratories can identify more strains of this species, its prevalence, disease spectrum, and clinical significance will remain in question. Moreover, it is not clear if the currently studied isolates cover the diversity of *E. albertii* as mostly only strains from three phylogroups have been analyzed. Therefore, a more diverse strain collection needs to be included in future studies. Care must also be taken that such characterization studies are carried out using diverse strains not multiple isolates of the same strain. For instance, it is not clear whether the reported 64 *E. albertii* strains isolated from 8 oyster samples [22] or 143 strains from 62 PCR-positive swab samples [49], represent multiple different strains or are simply multiple isolates of the same strain.

## 10. Conclusions

Although *E. albertii* strains continue to be misclassified, growing evidence indicates that this bacterium is an important human and animal pathogen. Much remains to be investigated regarding this pathogens’ virulence mechanisms and potentials, nutrient utilization, stress tolerance capacity, and their regulation. Our understanding of its pathobiology, as well as mechanisms of colonization, survival, and dissemination within and between hosts, is still limited and needs more analysis. There is a need for *E. albertii* surveillance, to establish the full extent of its public health significance and contributions to infectious diarrhea which would provide valuable knowledge for the development of intervention and control strategies. This will only be possible if simple and effective diagnostic tools are made available for its effective isolation and identification.

## Figures and Tables

**Figure 1 microorganisms-10-00875-f001:**
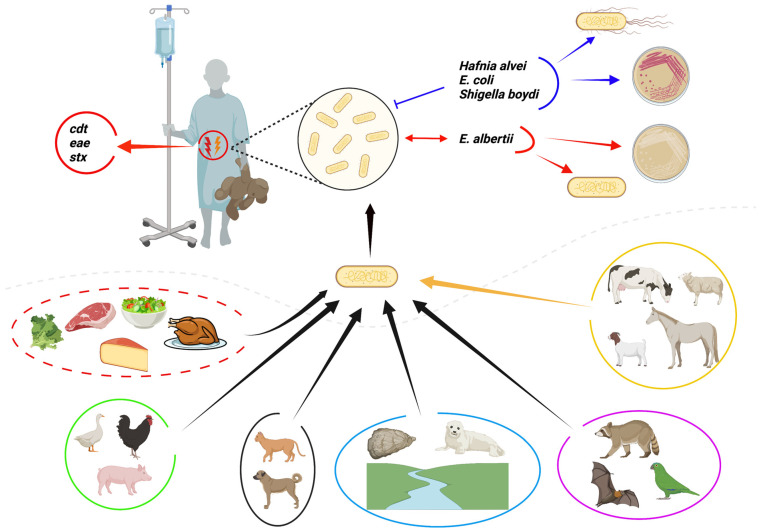
Overview of key aspects of *E. albertii*, an emerging elusive zoonotic foodborne pathogen. *E. albertii* infection mainly presents as watery diarrhea, with most cases being reported in children under 10 years. Virulence factors encoded by *cdtABC*, *eae*, *stx2a*, and *stx2f* seem to play a critical role in virulence. Blue arrows: *E. albertii* can be misclassified as *H. alvei*, *E. coli*, and *Shigella boydi.* Red arrows: based on *E. albertii* being non-motile and other phenotypes such as its inability to ferment rhamnose and xylose (white colonies) and the *Eacdt* and/or *EAKF1*_ch4033 gene presence, it can be differentiated from other similar species. Due to earlier reports that *E. albertii* cannot ferment lactose some lactose fermenting *E. albertii* strains might have been misidentified. Black arrows: confirmed sources of *E. albertii* include various foods with poultry products and water being chief culprits. Companion animals might also pose a risk due to close contact with humans while wild animals and birds are postulated to be potential reservoirs for this pathogen. Yellow arrow: although not yet isolated from livestock (cattle, sheep, goats, and horses), it is highly likely that they might also carry *E. albertii*. In support of this hypothesis, *E. albertii* has been isolated from mutton. Since several water sources have been confirmed to be contaminated with this pathogen, they pose a risk and can act as sources of contamination for seafood, aquatic animals, livestock, and vegetables when used for irrigation and drinking water without prior treatment. Figure created with BioRender.com.

**Figure 2 microorganisms-10-00875-f002:**
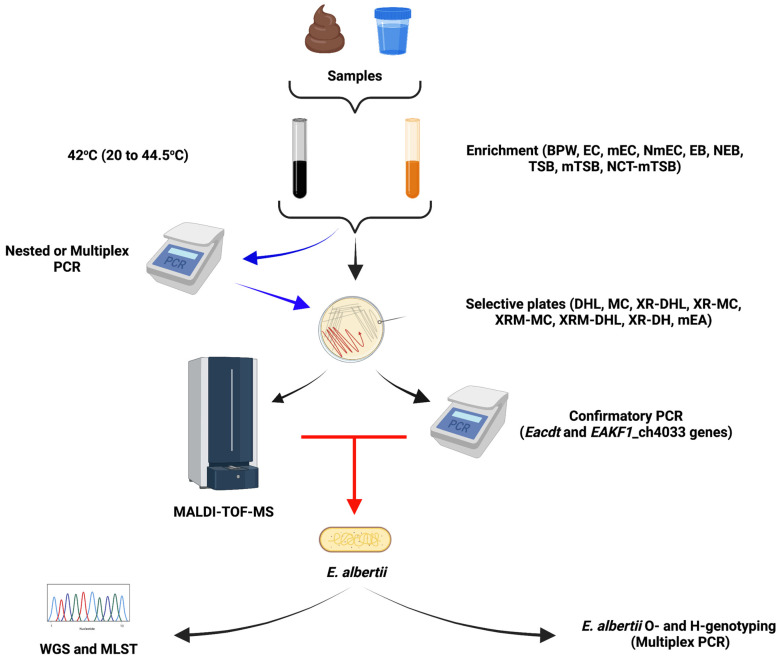
Compilation of described *E. albertii* enrichment and isolation protocols. Most protocols rely on a single enrichment step using various broth including buffered peptone water (BPW), *E. coli* broth (EC), modified EC (mEC), novobiocin–mEC (NmEC), *Enterobacteriaceae* enrichment broth (EB), novobiocin supplement EB (NEB), modified tryptic soy broth (mTSB), or novobiocin–cefixime–tellurite supplemented mTSB (NCT-mTSB) at various temperatures ranging from 20 to 44.5 °C. After this first step, most protocols recommend doing an *E. albertii* specific PCR (blue arrows) and only plating out enrichments with positive PCR results. While some protocols recommend direct plating of all enriched samples onto selective media and selecting for white colonies without doing an *E. albertii* specific PCR. On all media except XRM-MC, XR-DH, and mEA agar, lactose fermenting *E. albertii* might be missed through this approach, however, white colonies on these three plates might include other non-*E. albertii* species such as *Shigella* spp. Red arrow: Selected white colonies can be confirmed using MALDI-TOF-MS or *E. albertii* specific PCR targeting the *EAKF1*_ch4033 and *Eacdt* genes. However, for MALDI-TOF-MS a local *E. albertii* specific database library is required. Further strain characterization can be performed by WGS, MLST, H- and O-genotyping. Figure created with BioRender.com.

**Figure 3 microorganisms-10-00875-f003:**
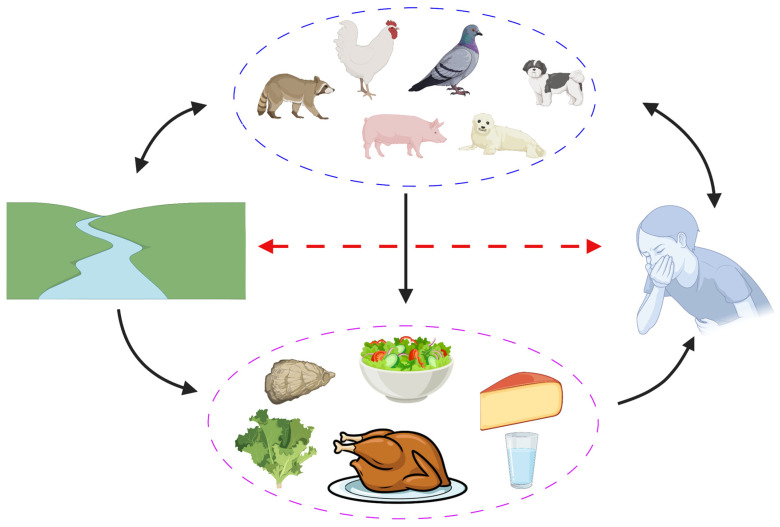
Potential transmission cycle of *E. albertii*. Animals act as a source and potential reservoir for *E. albertii*, either contaminating or being colonized/infected via water. Such water can contaminate plants or food and can be a direct source of *E. albertii* to humans. Although not confirmed, it is highly likely that ill individuals with diarrhea can contaminate the environment and water (red arrow), potentially contaminating food prepared under poor hygiene conditions. However, human *E. albertii* carriers have not yet been reported. Figure created with BioRender.com.

**Table 1 microorganisms-10-00875-t001:** Biochemical properties of *Escherichia albertii*.

Condition	Phenotype	Condition	Phenotype
Lactose	-v	Acid from: Glycerol *	+v
D-Xylose	-v	D-Glucose (+gas)	+
Melibiose	-v	Adonitol	-
Sucrose	-v	Amygdalin	-
L-Rhamnose	-v	D-Arabinose	v
D-Sorbitol	v	Indole	+v
D-Arabitol	-	Oxidase	-
Cellobiose **	-	Catalase	+
Acetate	+v	Voges-Proskauer (25 °C)	-
D-Mannitol	+v	Voges-Proskauer (35 °C)	-
Trehalose	+v	Lysine decarboxylase	+v
Maltose	v	Arginine dihydrolase (4 days)	-v
Lactulose	v	Ornithine decarboxylase (4 days)	+v
D-Arabinose	+v	Glutamate decarboxylase	+
Sedoheptulose anhydride	-	β-galactosidase	+v
D-Galactose	+	Pigment	-
D-Mannose	+	Urea	-
D-Ribose	+	MUG	-v
D-Fructose	+	PPA	-
Raffinose	-v	Methyl red	+
Salicin	-	Nitrate reductase	+
D-Lyxose	-	ONPG	+v
L-Arabinose	+v	Protease	-
Dulcitol	-	H_2_S production on: TSI GCF	--
Citrate	-v	Pectinase	-
D-Fucose	-	Degradation of: Mucate	-
Palatinose	-	Elastin	-
L-Sorbose	+v	Gelatin	-
D-Tagatose	-v	Hide powder	-
alpha-Methyl-D-glucoside	-	Polypectate (25 °C)	-
Erythritol	-	Tyrosine crystals	-
*i*-Inositol	-	DNA	-
D-Turanose,	-	Corn oil (lipase)	-
Xylitol	-	Hydrolysis of: Esculin	v
Malonate	-	Arbutin	-
Growth in KCN broth	-	L-Prolineaminopeptidase	v
3-Hydroxybenzoate assimilation	+	2-Ketogluconate assimilation	-
Myoinositol utilization	-	Histidine assimilation	-

Key: -: negative, +: positive, v: variable strain-dependent positive or negative phenotypes, -v: variable phenotype but most strains are negative, and +v: variable phenotype but most strains are positive. H_2_S: hydrogen sulphide; TSI: Triple sugar iron agar; GCF: gelatin-cysteine-thiosulfate medium; MUG: Methylumbelliferyl glucuronide cleavage by β-D-glucuronidase; ONPG: o-nitrophenyl-β-D-galactopyranoside; and PPA: phenylpyruvic acid. * Acid production from glycerol is a variable phenotype observed in some strains after prolonged incubation (3–7 days). ** Murakami et al. [36], reported a single strain that could utilize cellobiose. Table 1 is a combination of data derived from [1,2,10,22,26,31,32,33,35,36,37,38,40,41].

**Table 2 microorganisms-10-00875-t002:** *E. albertii* biogroups ^a^.

Biogroup	Indole Production	Lysine Decarboxylase	Acid Production from D-Sorbitol
1	-	+	+
2	+	-	-
3	+	+	NA

^a^ Adapted from Murakami et al. [36] and Nataro et al. [42]. *E. albertii* biogroup 1 and 2 described by Nataro et al. [42] also considered acid production from D-sorbitol, biogroup 1 are D-sorbitol-positive, while biogroup 2 are D-sorbitol-negative. Key: -: negative, +: positive, NA: not applicable.

**Table 3 microorganisms-10-00875-t003:** In-silico PCR analysis.

Gene	Primers ^a^	Total Tested ^b^	*E. albertii*	Missed ^c^	non-*E. albertii* ^d^
*Eacdt*	fw: GCTTAACTGGATGATTCTTG rv: CTATTTCCCATCCAATAGTCT	319	310	6	3
*EAKF1*_ch4033	fw: GTAAATAATGCTGGTCAGACGTTA rv: AGTGTAGAGTATATTGGCAACTTC	319	305	11	3
**Gene**	**Primers ^a^**	**Total Tested ^e^**	** *E. albertii* **	**Missed ^c^**	**non-*E. alberti* ^f^**
*Eacdt*	fw: GCTTAACTGGATGATTCTTG rv: CTATTTCCCATCCAATAGTCT	178	20	6	1
*EAKF1*_ch4033	fw: GTAAATAATGCTGGTCAGACGTTA rv: AGTGTAGAGTATATTGGCAACTTC	178	25	1	0

^a^ Adapted from Hinenoya et al. [24] and Lindsey et al. [51]. ^b,e^ WGS available on NCBI. ^c^ True *E. albertii* not picked up by the tested primers. ^d^ Reclassified as *E. ruysiae.* ^f^ Strain positive on the *Eacdt* or *EAKF1*_ch4033 PCR but correctly classified as *E. coli*.

**Table 4 microorganisms-10-00875-t004:** Major virulence factors of *E. albertii* ^a^.

Gene	Function or Annotation	Reference
LEE	Locus of enterocyte effacement	[5,13,78]
*eae*	Intimin; formation of attaching-effacing lesions	[5,13,78]
ETT2	*E. coli* type III secretion system 2	[13,14]
*paa*	Porcine attaching-effacing associated protein	[90]
*cdtABC*	Cytolethal distending toxin	[5,6,91]
*stx*	Shiga toxin	[5,6,13,39,45,91,97,99]
*hlyABCD*	Cytotoxicity	[6]
*iuc-ABCD*	Iron acquisition and transport	[6]
*ent* and *fep*	Enterobactin synthesis/iron acquisition	[48,67]

^a^ Functions of some of these virulence factors have not yet been experimentally confirmed.

## Data Availability

Not applicable.

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
