# Peer review of "Microbiology and Epidemiology of *Escherichia albertii*—An Emerging Elusive Foodborne Pathogen"

_microorganisms, 2022, doi:10.3390/microorganisms10050875_

Round 1

Reviewer 1 Report

the manuscript covers an important topic related to the emergency of a new Escherichia pathogen.

It is original since it collects all the information related to diagnosis and epidemiology filling the gap in a new and emerging topic

the figures are self explicative and well prepared

the manuscript is well written and the arguments well discussed

Maybe, the authors might consider to add some tables summarizing main content of paragraph. as for instance for paragraph 6 and 7

Reviewer 2 Report

This is a very interesting review on a still rather unknown opportunistic pathogen. I think there are some factors that could help to improve the review still a little bit. On the one hand I would introduce a section on Genomic Characterization of E. albertii. On the other hand, in the section that discusses methodologies that can be used to characterize this bacterium and aid in its identification, the authors could make some reference to the use of Whole Genome Sequencing. It is still an expensive technology that cannot be implemented in all laboratories. But it can be very useful, especially in the case of food outbreaks. More so in a pathogen that is difficult to classify using traditional methods.

Other few comments:

Line 184: “Several PCR protocols”.

Figure 2 maybe could be redesigned. It is a bit difficult to understand. For example, after enrichment. The authors could put two black arrows, one directed to a picture of a PCR and the other one the agar plates. Maybe with a arrow interconnecting both explaining when do one or the other. Then, also indicate with black arrow that it is possible to follow to ways. One PCR and the other one Maldi-TOF. After that the authors add an image of a bacteria with the name of E. albertii.

This recent reference should be included:

Wang, H., Zhang, L., Cao, L., Zeng, X., Gillespie, B., & Lin, J. (2022). Isolation and characterization of Escherichia albertii originated from the broiler farms in Mississippi and Alabama. Veterinary Microbiology, 267, 109379.

Round 2

Reviewer 2 Report

The authors responded to all my queries